# Water Resilience by Centipedegrass Green Roof: A Case Study

**Shuai Hu, Lijiao Liu, Junjun Cao, Nan Chen and Zhaolong Wang ***

School of Agriculture and Biology, Shanghai Jiaotong University, Shanghai 200240, China; hu851860902@sjtu.edu.cn (S.H.); 2234255580@situ.edu.cn (L.L.); junjunstar@sjtu.edu.cn (J.C.); chennnn@sjtu.edu.cn (N.C.)

* Correspondence: turf@sjtu.edu.cn

**Abstract:** Centipedegrass (*Eremochloa ophiuroides*) is a low-maintenance turfgrass. The first extensive green roof of centipedegrass was established in TongZhou Civil Squares in 2014. However, storm-water-runoff reduction, water-retention capacity, and plant-water requirements by a centipedegrass green roof has not yet been defined. The soil moisture dynamics, rainwater-retention capacity, runoff reduction, and plant evapotranspiration were investigated by simulated centipedegrass green roof plots, which were constructed in the same manner as the green roofs in TongZhou Civil Squares in 2018. The results showed that the centipedegrass green roof retained 705.54 mm of rainwater, which consisted 47.4% of runoff reduction. The saturated soil moisture was 33.4 ± 0.6%; the excess rainfall over the saturated soil moisture resulted in runoff. The capacity of rainwater retention was negatively related to the soil moisture before rain events and was driven by plant evapotranspiration. Drought symptoms only occurred three times over the course of a year when the soil moisture dropped down to 10.97%. Our results indicate that the rainwater retained in the soil almost met the needs of plant consumption; a further increase of rainwater retention capacity might achieve an irrigation-free design in a centipedegrass green roof.

**Keywords:** green roof; centipedegrass; runoff reduction; evapotranspiration; irrigation-free design

## 1. Introduction

Floods are natural disasters and cause over one-third of global economic losses according to the report of United Nations [1]. Flood disasters occur more frequently in urban areas because most urban surfaces are impervious roofs and roads [2,3].

Nature-based solutions (NBS) use nature and ecosystem services to provide social, economic, and environmental benefits [4,5]. One NBS is a green roof, which could provide benefits of stormwater management and urban heat mitigation [6–8] without occupying precious and highly competitive land at ground level [9]. Benefits of green roofs are mainly achieved by the performance of plants; however, the shallow substrates with limited availability of water and nutrients present a great challenge for the growth and survival of green roof plants [10,11]. Therefore, plant species with low-water requirements and the high-water use efficiency were selected to be used on the extensive green roofs [12,13]. Sedum plants have attracted considerable attention for use in green roofs because of their superior drought tolerance and can survive without irrigation in most areas [14–16].

The capacity of runoff reduction in a green roof was determined by substrate water content before a rainfall event, the size of a rainfall event, and the amount of water lost between two rain events via evapotranspiration [17–20]. Plant species with lower transpiration rates resulted in the less water loss and smaller volume of rainwater retention, less runoff reduction, and offered a minor contribution to storm water management [21,22]. Higher evapotranspiration of the extensive green roof could be achieved by using plant species with high transpiration rates [23], but frequent irrigation may

be required to supply the water consumption by plants [24], which increased maintenance cost and limited its commercial implementation [25,26].

Recent studies have found that increased diversity of plant species can provide multiple environmental benefits [27–30]. Ntoulas et al. [31] reported that three warm-season turfgrass species could perform well in the extensive green roofs with 15 cm depth of substrate. Speak et al. [32] found that turfgrass species were more effective at PM10 capture than sedums. Our previous study also found that turfgrass species performed better in runoff reduction than sedums [9].

Centipedegrass (*Eremochloa ophiuroides*) is a low-maintenance, warm-season turfgrass species [33]. Our previous studies found that centipedegrass not only served as environmental phytoremediation [34] but also performed better summer cooling than sedums when it was planted in extensive green roofs [35]. A centipedegrass green roof (12,000 m$^2$) was successfully established on the five conjoined buildings in TongZhou Civil Squares in 2014 and has become a popular place of leisure for local residents. Only two to three irrigation operations have been necessary each year because centipedegrass did not show any drought-stress symptoms during most periods of the year. The characteristics of water resilience by a centipedegrass green roof have not yet been well defined. Therefore, the objectives of this study were: (1) to investigate the dynamics of soil moisture and its runoff reduction function; (2) the relationships among water retention, plant evapotranspiration, and irrigation requirement; and (3) perspective to achieve a balance of plant water consumption and rainwater retention.

## 2. Materials and Methods

### 2.1. Centipedegrass Extensive Green Roof Project

The centipedegrass green roof was established in 2014 with a total area of 12,000 m$^2$ on the building roofs of Civil Squares in TongZhou District (32°07′07.39″ N, 121°08′35.7″ E), which includes five buildings: the Citizen Service Center, Women and Children Center, Culture Center, Archive, and Library, as seen in Figure 1A. All green roofs are accessible to the public, with the roof slope angle varying from 2% to 27%, as seen in Figure 1B,C. The extensive green roof was constructed with a waterproof layer, 24 cm plant growth substrate of the locate sandy soil, and centipedegrass (*Eremochloa ophiuroides* cv. 'Civil') on the flat roofs, and with additional antiskid grid in the soil layer (polyethylene, with grid of 400 mm × 400 mm) on the roofs over 10% of the slope.

The green roofs were maintained extensively according to a minimum schedule, which included two mowings and two fertilizations per year since it was established. The first mowing (60 mm height) occurred at the end of March to remove the dead leaves and promote spring Greenup. The first fertilization followed the first mowing, with 20 g m$^{-2}$ of slow-released granular fertilizer (N:P$_2$O$_5$:K$_2$O = 16:16:16). The second mowing (100 mm height) occurred at the end of July during the peak growth period. The second fertilization period with 10 g m$^{-2}$ of slow-released granular fertilizer (N:P$_2$O$_5$:K$_2$O = 16:16:16) was in early October to promote fall growth and winter performance.

The extra irrigation system (TORO TMC-212, CA, USA) was constructed for the green roof. However, only two to three periods of irrigation have been undertaken on a yearly basis because centipedegrass did not show drought-stress symptoms during most periods of the year. Thus, we conducted a simulated plot experiment on the hydrological dynamics to define the water resiliency of the extensive centipedegrass green roof.

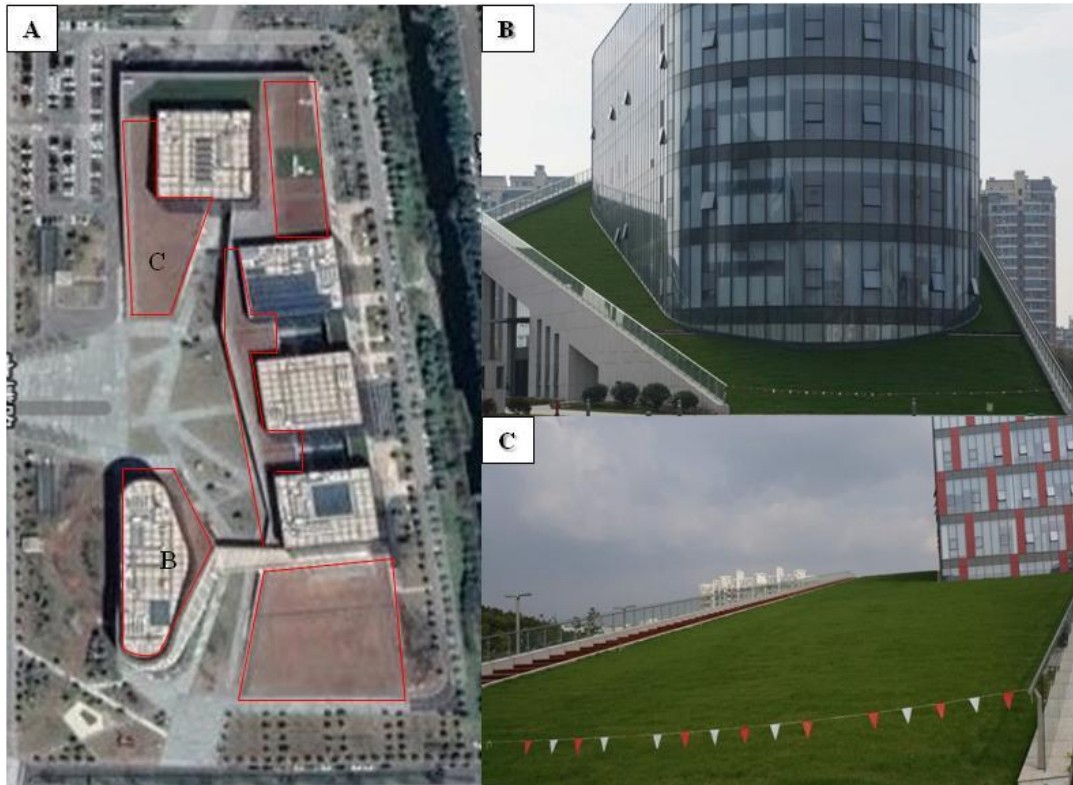

**Figure 1.** Photos of the green roof project. (**A**): Map of TongZhou Civil Squares; the red lines indicate the green roofs constructed using centipedegrass; (**B**): Slope green roof (27%) on the building of the Culture Center; (**C**): Flat green roofs on the buildings of Women and Children Center, Archive, and Library and a slope green roof on the building of Citizen Service Center.

### 2.2. Plot Experiment

Four centipedegrass green roof plots with internal dimensions of 76 cm long × 36.5 cm wide × 24 cm height were constructed in the campus of Shanghai Jiaotong University to investigate the water resiliency of the extensive centipedegrass green roof. The green roof plot was the same as the flat green roof of TongZhou Civil Squares, with a waterproof layer, 24 cm of sandy soil rootzone, and centipedegrass (*Eremochloa ophiuroides* cv. 'Civil') on 2% of the slope. Each green roof plot was in a rectangular, high-density polyethylene plastic lysimeters with an outflow opening (1 cm in diameter) constructed at the lowest part of the lysimeters. The experimental green roof plots were maintained in the same manner as the green roof in TongZhou Civil Squares.

The experiment was randomly arranged with four replications. All measurements were conducted daily from January 1 to December 31 2018. ①Air temperature was measured 1 m above the green roof surface by a thermometer (Shenzhen Tuo Er Wei Electronic Technology Co., Ltd., China). Rainfall was monitored by a rain gauge during the rain event. ②The dynamics of the soil moisture was monitored by weight of the green roof plot, which is calculated by:

Soil moisture (%) = (the readings of soil weight − the soil dry mass)/the soil dry mass × 100%

where, the daily soil weight = the weight of green roof − the weight of plant. The weight of plant was evaluated by seasonal samplings from a monitored column in the experimental plots.

③The measurement of evapotranspiration (ET) was performed according to our previous study [9]. The mass of the green roof plot was recorded daily and the mass changes represent the water loss by ET of the green roofs. ④Water retention of the green roof was calculated by the weight increase of the green roof plot in each raining event. ⑤Runoff was collected and recorded by the outflow of each green roof lysimeter during the raining days. Runoff reduction was calculated by:

Runoff reduction (%) = (Runoff on the cement roof − runoff of the green roof)/Runoff on the cement roof × 100%

### 2.3. Statistical Analysis

All data are presented as means of four replications. Statistical analysis was performed with the software SAS (version 9.1, SAS Institute Inc., Cary, NC) using the regression models. Correlation significance was tested by least significance difference (LSD) at a 0.05 probability level.

## 3. Results

### 3.1. Subsection

#### 3.1.1. Soil Moisture Dynamics

Soil moisture of the green roof was decreased by evapotranspiration (ET) water loss during the dry days and increased by the rainwater infiltrated into the soil during rainy days. The yearly soil moisture dynamics is presented in Figure 2A. The field capacity of centipedegrass green roofs was 33.4 ± 0.6%, shown as a blue dashed line, as seen in Figure 2A. When soil moisture reached its field capacity, runoff occurred from the outflow of the green roof plots. Soil moisture decreased gradually by daily water consumption of plants, represented as ET. Plant-leaf wilting symptoms occurred when the soil moisture dropped to a threshold of 10.97%, shown as a red dashed line, as seen in Figure 2A. Plant-leaf wilting only occurred three times: on June 9, July 16, and August 12. Two irrigations were applied to green roof plots with 52.05 mm on June 9 and 59.07 mm on July 16 to encourage plant recovery and avoid additional drought stress. No irrigation was applied on August 12 because 78 mm of heavy rain occurred on August 13, as seen in Figure 2B, and the plants soon recovered after the rain.

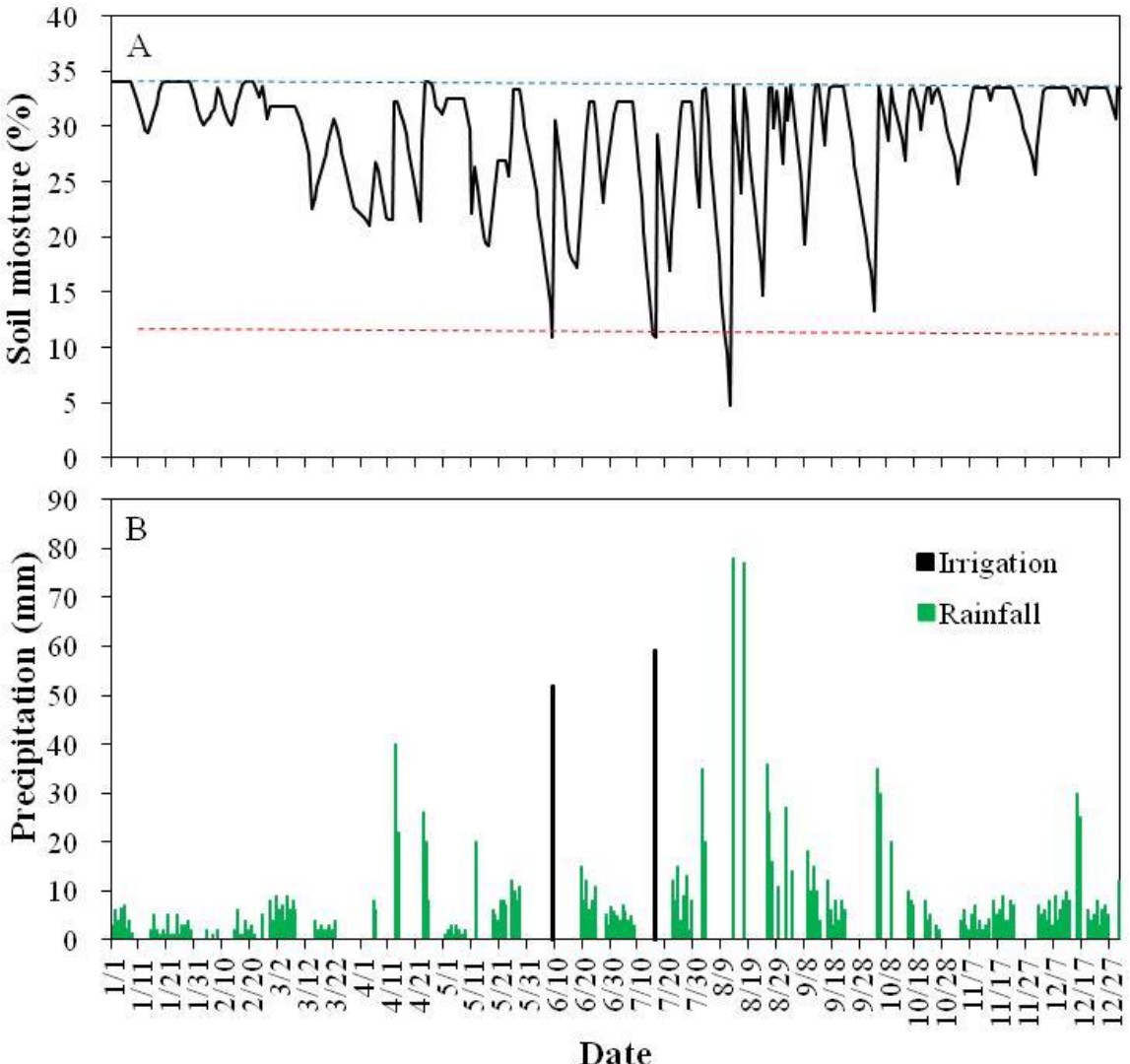

**Figure 2.** Soil moisture dynamics of centipedegrass green roof during 2018. (**A**) Soil moisture of the green roof. Blue dashed line represents the field capacity and the red dashed line represents the wilting point. (**B**) Rainfalls and irrigations to the green roof.

### 3.1.2. Runoff Reduction

There were 37 rain events and 33 runoffs over the course of 2018. Total amount of runoff was 781.56 mm, which was calculated as 47.4% of runoff reduction in 1487.1 mm of annual precipitation, as seen in Table 1. Full runoff reduction (in which no runoff occurred) occurred only four times, which was 5 mm of precipitation from February 3–8, 5 mm on February 24, 22 mm from March 15–22, and 14 mm from April 5–6, respectively. There were four serious runoffs, which occurred with 63.6mm (5.1% runoff reduction) from February 27–March 8, 52.2 mm (32.2% runoff reduction) on August 17, 57.8 mm (25.9% runoff reduction) from December 1–12, and 51.2 mm (6.9% runoff reduction) from December 15–16, respectively.

**Table 1.** Rainwater runoff reduction on the centipedegrass green roof.

| Date | Rainfall (mm) | Runoff (mm) | Runoff Reduction % | Date | Rainfall (mm) | Runoff (mm) | Runoff Reduction % |
|---|---|---|---|---|---|---|---|
| 1/1-1/8 | 34.0 | 34.0 | 0.0 | 8/17 | 77.0 | **52.2** | 32.2 |
| 1/15-1/29 | 38.0 | 25.0 | 34.2 | 8/25-8/27 | 78.0 | 29.0 | 62.8 |
| 2/3-2/8 | 5.0 | 0.0 | 100.0 | 8/29 | 11.0 | 2.5 | 77.4 |
| 2/14-2/21 | 20.0 | 9.1 | 54.3 | 9/1 | 27.0 | 8.8 | 67.6 |
| 2/24 | 5.0 | 0.0 | 100.0 | 9/3 | 14.0 | 5.8 | 58.9 |
| 2/27-3/8 | 67.0 | **63.6** | 5.1 | 9/9-9/13 | 57.0 | 19.3 | 66.1 |
| 3/15-3/22 | 22.0 | 0.0 | 100.0 | 9/16-9/22 | 47.0 | 32.9 | 29.9 |
| 4/5-4/6 | 14.0 | 0.0 | 100.0 | 10/4-10/5 | 65.0 | 12.0 | 81.5 |
| 4/13-4/14 | 62.0 | 35.7 | 42.5 | 10/9 | 20.0 | 7.5 | 62.6 |
| 4/23-4/25 | 54.0 | 18.0 | 66.6 | 10/15-10/17 | 25.0 | 7.8 | 69.0 |
| 5/1-5/8 | 15.0 | 11.5 | 23.1 | 10/21-10/23 | 17.0 | 7.4 | 56.8 |
| 5/12 | 20.0 | 9.6 | 51.9 | 10/25-10/26 | 5.0 | 1.6 | 67.4 |
| 5/18-5/23 | 38.0 | 2.0 | 94.7 | 11/3-11/13 | 42.1 | 19.4 | 53.8 |
| 5/25-5/28 | 41.1 | 39.0 | 5.1 | 11/15-11/22 | 51.0 | 48.2 | 5.6 |
| 6/19-6/24 | 60.0 | 22.9 | 61.8 | 12/1-12/12 | 78.0 | **57.8** | 25.9 |
| 6/28-7/8 | 54.0 | 31.4 | 41.9 | 12/15-12/16 | 55.0 | **51.2** | 6.9 |
| 7/22-7/29 | 71.0 | 33.4 | 53.0 | 12/19-12/26 | 44.0 | 40.1 | 8.9 |
| 8/2-8/3 | 55.0 | 26.6 | 51.7 | 12/30-12/31 | 21.0 | 13.9 | 33.9 |
| 8/13 | 78.0 | 2.5 | 96.8 | | | | |
| Annual | 1487.1 | 781.56 | **47.4** | | | | |

### 3.1.3. Rainwater Retention in Response to Soil Moisture

The retention capacity of rainwater was significant negatively in correlation to the soil moisture when exposed to rainfall, as seen in Figure 3.

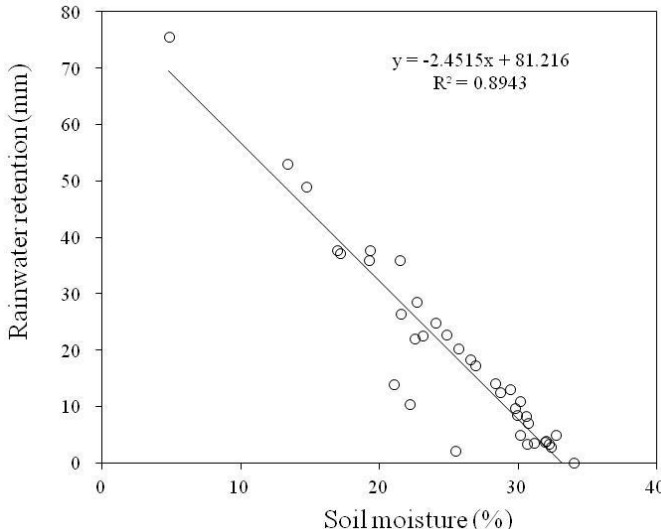

**Figure 3.** Green roof rainwater retention in response to soil moisture. It can be described as a linear equation: y = 81.216 − 2.4515x, with $R^2$ = 0.8943 **. In contrast: y = the amount of rainwater retention by the green roof; x = soil moisture before rainfall.

### 3.1.4. Evapotranspiration (ET) dynamics

The dynamics of green roof ET during 2018 is shown in Figure 4. The highest ET occurred in the summer months, with the average daily ET of 7.68 mm from July 10 to August 10. The lowest ET occurred in the winter months with only 2.14 mm and 2.07 mm in January and December, respectively. The green roof ET was significantly positively correlated to the air temperature, as seen in Figure 5.

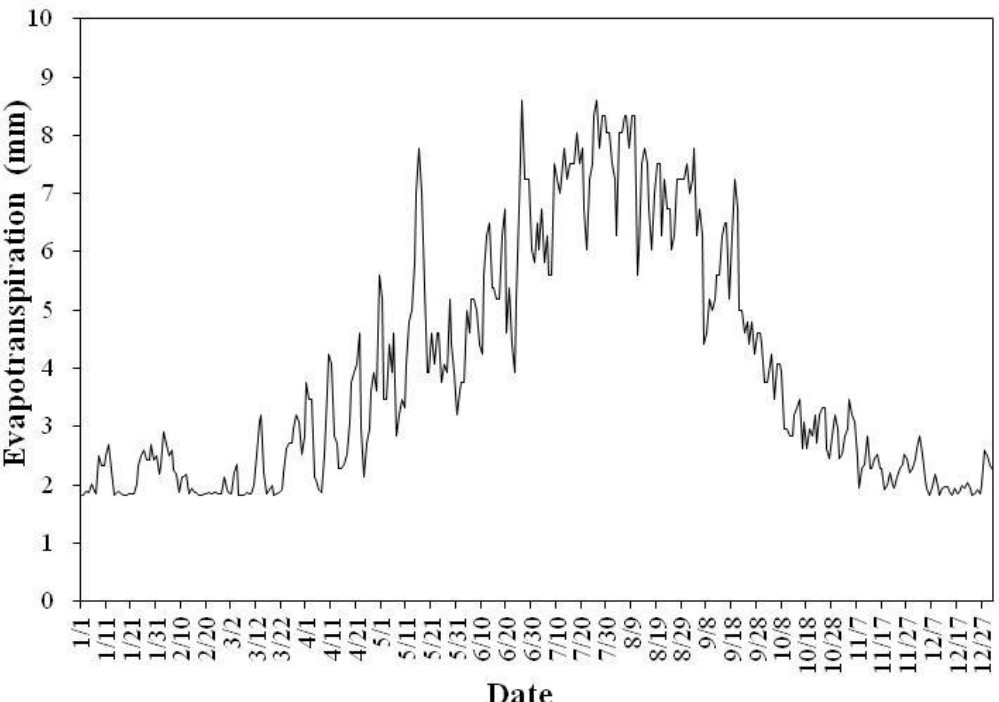

**Figure 4.** Evapotranspiration dynamics of centipedegrass green roof.

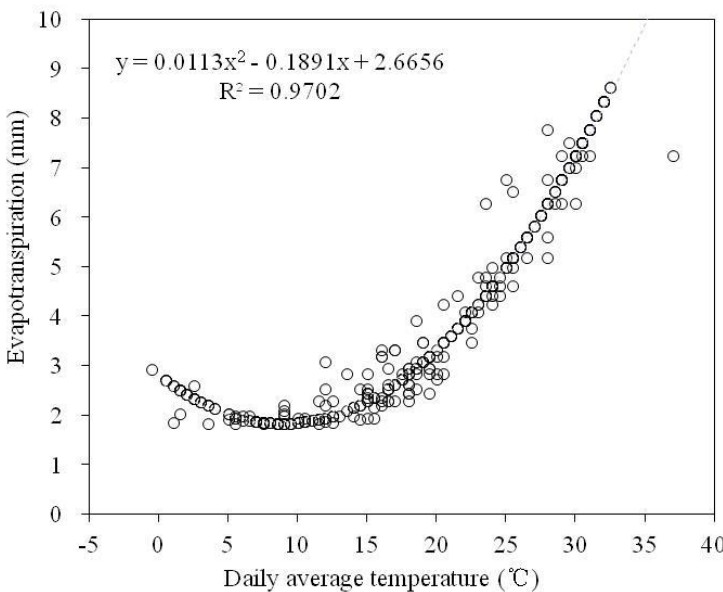

$$y = 0.0113x^2 - 0.1891x + 2.6656$$
$$R^2 = 0.9702$$

**Figure 5.** Green roof evapotranspiration in response to daily average temperatures. It can be described as a mathematical equation: $y = 2.6656 - 0.1891x + 0.0113x^2$, with $R^2 = 0.9702$ **. In contrast: y = green roof ET; x = daily average air temperature.

## 4. Discussion

The green building standard requires over 70% of runoff reduction, according to Chinese national standard 'Assessment Standard for Green Building, GBT53078-2014' [36]. The centipedegrass green roof only performed 47.4% of runoff reduction under the 1487.1 mm of annual precipitation, as seen in Table 1. Runoff reduction could be promoted by the increase of water retention capacity of the green roofs [9]. The field capacity represents the maximum rainwater retention of the green roof. In this study, the field capacity of centipedegrass green roofs was 33.4 ± 0.6%, as seen in Figure 2A. There are several ways to increase the field capacity of growth substrates. Soil physical and chemical properties could be improved by different soil amendments [37–40]. Water retention capacity and available water content could be greatly improved by organic manures [37], organic waste recycled in agriculture [38], Zeolites [39], the use of different types of biochar [40–42], or synthetic hydrogels [43,44]. Constructed green roofs with a deeper substrate layer could also retain more water and reduce runoff volume [45–47]. However, this option may not be applicable because the most extensive green roof has strict load limits [48,49]. Another option to enhance the runoff reduction and water retention capacity is to construct an additional water storage layer in the green roof system [19,50].

The capacity of rainwater retention by the green roof is also dependent on the dynamic changes of water consumption between two rainfall events [18], the period of dry days, and plant drought tolerance [51]. Water loss on green roofs was mainly driven by plant ET [52]. The more ET water lost, the more water storage space created, which means more water can be retained in the green roof system and the less runoff will be occurred during the next rainfall event [53].

The ET of centipedegrass green roof was significantly correlated to the air temperature with the seasonal changes, as seen in Figure 5. Most previous studies indicated that ET could be estimated and forecasted according to changes in temperature [54,55]. Water consumption by plant ET also creates a risk of plant-drought stress (i.e., leaf wilting) if the soil moisture dropped below the threshold [56]. The soil moisture threshold of the wilting point of centipedegrass is 10.97% which was only reached three times in 2018.

Extensive green roofs are designed for minimal maintenance with a shallow substrate layer [13]. No irrigation is required if the retained rainwater in the green roof system meets the plant water requirement during the dry period. In the current centipedegrass green roof system, only two irrigation events were applied. Increase of rainwater retention capacity could reduce or eliminate the need for irrigation. From the current data, an additional 39 mm rainwater retention capacity in the substrate would compensate for the water deficiency in all three dry periods (May 29–June 19, July 9–22, and August 4–13), and reach the water balance of rainwater retention and plant consumption, assuming that weather conditions, rainfall events, and plant ET do not vary each year.

The centipedegrass green roof in TongZhou Civil Squares provides not only environmental improvement but also an open leisure space for local residents. Plant performance was deemed satisfactory by the local government, constructor, maintenance manager, as well as the public in the first five years of its establishment. This research offers further improvement to achieve the balance of water retention and plant water requirements, which could result an irrigation-free design in the centipedegrass green roof.

## 5. Conclusions

Centipedegrass green roof reduced 47.4% of rainwater runoff under the 1487.1 mm of annual precipitation in 2018, with a field capacity of 33.4 ± 0.6%. The capacity of rainwater retention was linearly related to the soil moisture before rain events, which was driven by evapotranspiration of centipedegrass.

The rainwater retained in the current centipedegrass green roof system almost met the requirements for plant water consumption most of the year, except for three drought periods in the summer months. Irrigation of centipedegrass was required when soil moisture dropped to its wilting point of 10.97%. Our data showed that an additional 39 mm rainwater retention capacity would compensate for the

water deficiency in all three drought periods; it would also reach the water balance of rainwater retention and plant consumption, indicating that an irrigation-free design is possible through the increase of water retention capacity to achieve water balance in a centipedegrass green roof.

Centipedegrass is a warm-season turfgrass species and could be widely planted in most tropical and subtropical areas. Our results showed that it could be an option for extensive green roofs because it has been well adapted to a low maintenance schedule. Centipedegrass green roofs can provide a leisure place for local residents that sedum green roofs cannot.

**Author Contributions:** Conceptualization, Z.W., and S.H.; Methodology, J.C., and S.H.; Investigation and data analysis, J.C., S.H., L.L., and N.C.; Manuscript preparation and writing, Z.W., and S.H.

**Funding:** This research was funded by "National Natural Science Foundation of China, grant number 31872412" and "National Key Research and Development Program of China, grant number 2018YFD0800205".

**Conflicts of Interest:** The authors declare no conflict of interest.

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
