# Peer review of "Water Resilience by Centipedegrass Green Roof: A Case Study"

_buildings, doi:10.3390/buildings9060141_

Reviewer 1 Report

The comments concerns manuscript „Water resilience by centipedegrass green roof: A case study”

The manuscript refers to the important issue of active biological surface in cities. Green roofs are one of the remedies for these problems and the research subject taken is very important due to adaptations of cities to climate change.

However, the manuscript has many missing points and at the moment is not acceptable to be published in a such form.

General comments:

The manuscript is a total mess! Look at the author requirement at the Buildings website!  There should be:  Introduction, Materials and Methods, Results, Discussion, Conclusions. In your manuscript the methodology is almost at the end! Correct it!

Materials and Methods

There are a lot of missing gaps! First of all, why there is a description of real green roof (4.1), if you conducted your research at completely different place and as a plots of 76 x 36.5 x 24 cm… How many replications do you have? Instead of fig 5 with building, you should show the photo of your experiment!

There is lack of information about the climate, water conditions, precipitation and so on of the place where the experiment was conducted.

It should be detailed descripted how long the experiment lasts (how many days!), what measurements were done and so on. You have such information at the moment but is not sufficient!

What about statistics??? There is no information about this!

How the soil moistures were monitored?? By weight…??? What it the biomass of centipedegrass that could increase??? Did you thought about it? You should measure it by hygrometer or some wet sensor!

The is also no information about the centipedegrass. What about its water requirements??

l.79 part of methodology

Results

2.1.2 most of the information is repeated from the table 1! The result description should complete the data from the figures / table!

The description of plots are rather obvious… please delete them! (l. 90-91; 97-98)

What ** indicate?? No description both in methodology and results….

Discussion

l.119-120 repeating the results…

l. 126 references needed!

l.139-147  these are the results, not the discussion…

The conclusion is too general and need further refinement.

SI unites are obliged!

l.166 is this a symbol of slope….? Please correct it! Check it in other places!

l. 147 upper index

l. 172, 175, 177 add the year

At the moment, the manuscript is not comprehensive, underdeveloped and should be strictly rewritten!

Author Response

Manuscript ID: buildings-524339

Title: Water resilience by centipedegrass green roof: A case study

Journal: Buildings

Dear reviewer,

We appreciate you very much for the constructive comments and suggestions and giving us an opportunity to revise our manuscript. According to the detailed suggestions from you, we have made a careful revision on the original manuscript. All changes are highlighted using "Track Changes" in the revision file.

We hope that the revision is acceptable and look forward to hearing from you soon.

Best Regards,

Zhaolong Wang, Ph.D. Professor

The following is a point-to-point response to the reviewers’ comments.

The manuscript refers to the important issue of active biological surface in cities. Green roofs are one of the remedies for these problems and the research subject taken is very important due to adaptations of cities to climate change.

However, the manuscript has many missing points and at the moment is not acceptable to be published in a such form.

Answer: Thanks a lot for your comment.

General comments:

The manuscript is a total mess! Look at the author requirement at the Buildings website!  There should be:  Introduction, Materials and Methods, Results, Discussion, Conclusions. In your manuscript the methodology is almost at the end! Correct it!

Answer: Thanks a lot for your comment.

We received a Template from Guest Editor Prof. Dr. Manfred Köhler of Buildings, in which the order of manuscript is: 1. Introduction, 2. Results, 3. Discussion, 4. Materials and Methods, 5. Conclusions. We do not know if this order should be applied to Special Issue "Quantification of Green Roof Benefits and the Implementation into Urban Politics".

According to your comment, we changed the manuscript order as: Introduction, Materials and Methods, Results, Discussion, Conclusions.

Materials and Methods

There are a lot of missing gaps! First of all, why there is a description of real green roof (4.1), if you conducted your research at completely different place and as a plots of 76 x 36.5 x 24 cm…

Answer: We apologize for our unclear description and make you misunderstanding.

The experimental green roof plots was simulated to investigate the water resilience of this real green roof. The reasons for conducting this simulated plot experiment instead of the direct study in the real green roof are:

①The real green roof is a finished project. It is not allowed to destroy its integrity. It is impossible to take plots out for experimental measurements.

②The real green roof showed great water resilience after its established in 2014 because it only need minor irrigation during the maintenance. However, no data to qualify its hydrological performance.

③The simulated experimental plots were constructed extra the same as the real green roof and the study is conducted in the same climate zone with almost the same weather conditions.

How many replications do you have? Instead of fig 5 with building, you should show the photo of your experiment!

Answer: The replication of the simulated experimental plots was four. We added a sentance in Line 94: " The experiment was randomly arranged with 4 replications."

As described above, the purpose of this study is to qualify the real centipedegrass green roof, not a simulated experimental plot. So we showed the photo of real green roof. If the reviewer think it is not appropriate to show this real green roof. we will delete the photos.

There is lack of information about the climate, water conditions, precipitation and so on of the place where the experiment was conducted.

Answer: Thanks a lot for your comment.

Precipitation data (daily) is in Figure 1 and all rainfall events (period) is in Table 1.

Daily temperature changes is shown in the following. We added it in the supplemental material.

Figure S1. temperature dynamics above the green roof during the year of 2018. 

It should be detailed descripted how long the experiment lasts (how many days!), what measurements were done and so on. You have such information at the moment but is not sufficient!

Answer: Thanks a lot for your comment.

The measurement was conducted in whole year of 2018 (Jan. 1st~Dec. 31th).

A sentence was added in Lines 94-96: "All measurements were conducted daily Temperature and rainfall were recorded daily during the whole year of 2018 (from January 1st to December 31th)."

The measurements were presented in Lines 96-105:

"①Air temperature was measured on 1 m above the green roof surface by a thermometer (Shenzhen Tuo Er Wei Electronic Technology Co., Ltd, China). Rainfall was monitored by a rain gauge during the raining day. ②The dynamic of the soil moisture was monitored by weight of the green roof plot, which is calculated by: Soil moisture (%) = (the readings of soil weight - the soil dry mass) / the soil dry mass × 100%   ③The measurement of evapotranspiration (ET) was performed according to our pervious study [9]. The mass of the green roof plot was recorded daily and the mass changes represent the water loss by ET of the green roofs. ④Water retention of the green roof was calculated by the weight increase of the green roof plot in each raining event. ⑤Runoff was collected and recorded by the outflow of each green roof lysimeter during the raining days. "

What about statistics??? There is no information about this!

Answer: Thanks a lot for your comment.

We added the description of statistics in Lines 108-111:

"2.3 Statistical Analysis

    All data are presented as means of four replications. Statistical analysis was performed with the software SAS (version 9.1, SAS Institute Inc., Cary, NC) using the regression models. Correlation significance was tested by Least significance difference (LSD) at a 0.05 probability level."

How the soil moistures were monitored?? By weight…??? What it the biomass of centipedegrass that could increase??? Did you thought about it? You should measure it by hygrometer or some wet sensor!

Answer: Thanks a lot for your comment.

Yes, we measured soil moisture by weight changes of green roof plots. We found that measuring soil moisture by weight is reliable and accurate in green roof study. Because:

① Plant biomass only consist 0.3~0.6 % of green roof weight.

② The growth of centipedegrass was relatively stable. The biomass changes by centipedegrass growth is minor under the extensive maintenance schedule. The biomass of centipedegrass can be estimated by sampling weight in each season.

③ The hydrometer reading is not as sensitive as the weight changes under the dry condition, especially when soil moisture dropped under 15%.

The is also no information about the centipedegrass. What about its water requirements??

Answer: Thanks a lot for your comment.

Centipedegrass is a warm-season turfgrass species. As we know, this is the first experiment to study its water requirement on green roof.

l.79 part of methodology

Answer: Thanks a lot for your comment. Rain events and runoff were the results of our measurements (Table 1)

Results

2.1.2 most of the information is repeated from the table 1! The result description should complete the data from the figures / table!

Answer: Thanks a lot for your comment.

Table 1 only presented the data of each rainfall events (periodically), runoffs during the rain events, and runoff reduction. Figure 1A presented the dynamics of soil moisture (daily changes), Figure 1B presented the daily precipitation and irrigation volumes (water input into the green roof). The daily precipitation was not the repeated information of periodically rainfall events in Table 1 (water input, soil retention, and output of the green roof).

The description of plots are rather obvious… please delete them! (l. 90-91; 97-98)

Answer: Thanks a lot for your comment.

Deleted.

What ** indicate?? No description both in methodology and results….

Answer: Thanks a lot for your comment. We rewrote the Conclusion.

Discussion

l.119-120 repeating the results…

Answer: Thanks a lot for your comment.

It was changed to: " The green building standard requires over 70% of runoff reduction, according to Chinese national standard 'Assessment Standard for Green Building, GBT53078-2014' [36]. The centipedegrass green roof retained 781.56 mm of rainwater annually, which only performed 47.4% of runoff reduction under the 1487.1 mm of annual precipitation (Table 1). The runoff reduction could be promoted by the increase of water retention capacity of the green roofs [9]. " (Lines: 172-178)

l. 126 references needed!

Answer: Thanks a lot for your comment.

Citations added: "Soil physical and chemical properties could be improved by the different soil amendments [37-40 ]". (Line 182)

l.139-147  these are the results, not the discussion…

Answer: Thanks a lot for your comment.

We changed into: " The soil moisture threshold of centipedegrass wilting point was 10.97% which was only reached 3 times in the year of 2018." (Lines 203-204)

"In the current centipedegrass green roof system, only 2 irrigation events were applied for the centipedegrass green roof. Increase of rainwater retention capacity could reduce or eliminate the need for irrigation. From the current data, additional of 39 mm rainwater retention capacity by the substrate would compensate the water deficiency in all three dry periods (May 29~June 19th, July 9~22th, and August 4~13th), and reach the water balance of rainwater retention and plant consumption, assuming that weather conditions, rainfall events, and plant ET are not changed year by year." Lines: 207-213.

The conclusion is too general and need further refinement.

Answer: Thanks a lot for your comment.

We rewrote the conclusion:

 "Centipedegrass green roof reduced 47.4% of rainwater runoff under the 1487.1 mm of annual precipitation in 2018, with the field capacity of 33.4±0.6%. The capacity of rainwater retention was linearly related to the soil moisture before the rain events, which was driven by evapotranspiration of centipedegrass.

 The rainwater retained in the current centipedegrass green roof system could almost meet the requirement of plant water consumption in the most time of the year, except for three drought period in the summer months. Irrigation of centipedegrass was required when the soil moisture dropped to its wilting point of 10.97%. Our data showed that additional of 39 mm rainwater retention capacity would compensate the water deficiency in all three drought periods and reach the water balance of rainwater retention and plant consumption, indicating that an irrigation-free design could be possible by the increase of water retention capacity to achieve the water balance in centipedegrass green roof.

 Centipedegrass is a warm-season turfgrass species and could be widely planted in most tropical and subtropical areas. Our results showed that it could be an option for the extensive green roofs because it has been well adapted to the low maintenance schedule. Centipedegrass green roof can provide a leisure place for local residents, but sedums green roof cannot."

SI unites are obliged!

l.166 is this a symbol of slope….? Please correct it! Check it in other places!

Answer: Thanks a lot for your comment.

Slope unite is changed to %.

l. 147 upper index

Answer: Thanks a lot for your comment.

Changed.

l. 172, 175, 177 add the year

Answer: Thanks a lot for your comment.

The maintenance schedule was the same in the year since it was established. We added "which include 2 mowings and 2 fertilizations per year since it was established". Line 74.

At the moment, the manuscript is not comprehensive, underdeveloped and should be strictly rewritten!

Answer: We hope that the revision is acceptable and look forward to hearing from your comments.

Reviewer 2 Report

Dear Authors,

after reading I found your paper interesting but it needing some improvements before publication. Research on rainwater management is obviously necessary. However, this study has little interest in the international scale, and very much focused on the specific case study. But the outcomes can be beneficial for designing local strategies.

The applied methods were professional and effective in attaining the object of this work. In my opinion the structure of article should be improved. The section "4. Materials and Methods" should be before the sections 2 and 3. The part “5. Conclusions” is the weakest section of this paper. Please use your results as a starting point for developing a conclusion section of wider interest for the global scientific community.

Author Response

Dear reviewer,

We appreciate you very much for the constructive comments and suggestions and giving us an opportunity to revise our manuscript. According to the detailed suggestions from you, we have made a careful revision on the original manuscript. All changes are highlighted using "Track Changes" in the revision file.

We hope that the revision is acceptable and look forward to hearing from you soon.

Best Regards,

Zhaolong Wang, Ph.D. Professor

The following is a point-to-point response to your comments.

Comments and Suggestions for Authors

after reading I found your paper interesting but it needing some improvements before publication. Research on rainwater management is obviously necessary. However, this study has little interest in the international scale, and very much focused on the specific case study. But the outcomes can be beneficial for designing local strategies.

Answer: Thanks a lot for your comment.

Yes, it was a local case study for the centipedegrass green roof. However, centipedegrass can be grown in all areas which is suitable for the warm-season turfgrasses in the world. Our results should attract broad interests by using centipedegrass extensive green roof to create a roof leisure place under the minimum maintenance schedule (sedums cannot).

The applied methods were professional and effective in attaining the object of this work. In my opinion the structure of article should be improved. The section "4. Materials and Methods" should be before the sections 2 and 3. The part “5. Conclusions” is the weakest section of this paper. Please use your results as a starting point for developing a conclusion section of wider interest for the global scientific community.

Answer: Thanks a lot for your comment.

The order of manuscript was changed: 1. Introduction, 2. Materials and Methods, 3. Results, 4. Discussion, 5. Conclusions.

We rewrote Conclusion as:

 "Centipedegrass green roof reduced 47.4% of rainwater runoff under the 1487.1 mm of annual precipitation in 2018, with the field capacity of 33.4±0.6%. The capacity of rainwater retention was linearly related to the soil moisture before the rain events, which was driven by evapotranspiration of centipedegrass.

 The rainwater retained in the current centipedegrass green roof system could almost meet the requirement of plant water consumption in the most time of the year, except for three drought period in the summer months. Irrigation of centipedegrass was required when the soil moisture dropped to its wilting point of 10.97%. Our data showed that additional of 39 mm rainwater retention capacity would compensate the water deficiency in all three drought periods and reach the water balance of rainwater retention and plant consumption, indicating that an irrigation-free design could be possible by the increase of water retention capacity to achieve the water balance in centipedegrass green roof.

 Centipedegrass is a warm-season turfgrass species and could be widely planted in most tropical and subtropical areas. Our results showed that it could be an option for the extensive green roofs because it has been well adapted to the low maintenance schedule. Centipedegrass green roof can provide a leisure place for local residents, but sedums green roof cannot."

Round  2

Reviewer 1 Report

Thank you for comprehensive work that improved the quality of the manuscript!

I have one more comment – Please add the explanation about the moisture measurement to the material and method part (l. 98-99) why you decided to use such method based on your explanation:

/Plant biomass only consist 0.3~0.6 % of green roof weight.

The growth of centipedegrass was relatively stable. The biomass changes by centipedegrass growth is minor under the extensive maintenance schedule. The biomass of centipedegrass can be estimated by sampling weight in each season./

Author Response

Dear Reviewer:

Thanks a lot for your comment. Here is our response:

I have one more comment – Please add the explanation about the moisture measurement to the material and method part (l. 98-99) why you decided to use such method based on your explanation

Answer: Thanks a lot for your suggestion. We added explanation of the soil moisture measurement in Materiala and Methods (Lines 101-102):

Soil moisture (%) = (the readings of soil weight - the soil dry mass) / the soil dry mass × 100%  

Where, The daily soil weight = the weight of green roof - the weight of plant. The weight of plant was evaluated by seasonal samplings from a monitored column in the experimental plots.